# Cost-Efficient Online Decision Making:
# A Combinatorial Multi-Armed Bandit Approach

**Arman Rahbar**[*]                                    *armanr@chalmers.se*
*Chalmers University of Technology and University of Gothenburg*

**Niklas Åkerblom**[*]                              *niklas.akerblom@volvocars.com*
*Volvo Car Corporation*
*Chalmers University of Technology and University of Gothenburg*

**Morteza Haghir Chehreghani**                    *morteza.chehreghani@chalmers.se*
*Chalmers University of Technology and University of Gothenburg*

**Reviewed on OpenReview:** *https://openreview.net/forum?id=vZGZIIgcG4*

## Abstract

Online decision making plays a crucial role in numerous real-world applications. In many scenarios, the decision is made based on performing a sequence of tests on the incoming data points. However, performing all tests can be expensive and is not always possible. In this paper, we provide a novel formulation of the online decision making problem based on combinatorial multi-armed bandits and take the (possibly stochastic) cost of performing tests into account. Based on this formulation, we provide a new framework for cost-efficient online decision making which can utilize posterior sampling or BayesUCB for exploration. We provide a theoretical analysis of Thompson Sampling for cost-efficient online decision making, and present various experimental results that demonstrate the applicability of our framework to real-world problems.

## 1 Introduction

*Online decision making* (ODM) is concerned with interactions of an agent with an environment through a series of sequential tests in order to acquire sufficient information about the environment and successively learn how to make better decisions. Each such test and decision can be associated with either a reward or a cost. Here, we focus on problem settings where tests may incur stochastic costs.

One such example is medical diagnosis. A patient arrives at a hospital with an unknown affliction. In order to determine an appropriate treatment for the patient, a number of medical tests need to be performed. Since each of these tests might take time (which can differ depending on the test result and the underlying affliction), be expensive or cause discomfort to the patient, it is desirable to limit the set of tests to the most informative ones. On the other hand, it is apparent that misdiagnosing the patient can have severe consequences, so performing too few tests is also undesirable.

Another example is adaptive identification of driver preferences for in-car navigation systems. Consider the selection of charging stations during long-distance trips with battery-electric vehicles (BEVs). Since the duration of a single charging stop can be as long as 40 minutes, it may be desirable to select a charging station with additional amenities in close proximity, such as restaurants, shops, or public restrooms. While it is possible to ask questions through the vehicle user interface of drivers to learn their charging location

---

[*]These authors contributed equally to this work.

preferences, such questions should be kept few, short, and simple. We note that in both of these examples, the problem instances (e.g., diagnosis of patients and identification of driver preferences) arrive incrementally in an *online* manner and should be processed accordingly.

We take a novel view of the online decision making problem described above, where we cast it as a *combinatorial multi-armed bandit problem* (CMAB) (Cesa-Bianchi & Lugosi, 2012), an extension of the classical *multi-armed bandit problem* (MAB). The standard MAB problem is a common way of representing the trade-off inherent to decision making problems between exploring an environment by performing successive actions (playing arms, in bandit nomenclature) to gain more knowledge and exploiting that knowledge to reach a long-term objective. In a CMAB problem, an agent interacts with the environment in each time step by performing a set of actions (a *super arm*), where the set may be subject to combinatorial constraints. Each individual action (*base arm*) in that set may result in some observed feedback (so-called *semi-bandit* feedback), and the agent also receives a reward associated with the complete super arm.

In this work, we view the combined choice of tests to perform and the selected decision as a super arm in a CMAB problem where the underlying cost distribution parameters are initially unknown, which allows us to leverage effective MAB exploration strategies like *Thompson Sampling* (TS) (Thompson, 1933) and *Upper Confidence Bound* (UCB) (Auer et al., 2002) together with cost-efficient (active) information acquisition methods, such as *Equivalent Class Edge Cutting* ($EC^2$) (Golovin et al., 2010) and *Information Gain* (IG) (Dasgupta, 2005). We summarize our contributions as follows.

1. We study the novel problem of outcome-dependent cost-efficient online decision making, where costs may be stochastic and depend on both test and decision outcomes.

2. We propose a novel combinatorial semi-bandit framework for our online decision making problem. This formulation involves an elegant definition of base arms and super arms, providing a new perspective on cost-efficient information acquisition while simultaneously exploring the concept of interactive CMAB oracles.

3. We adapt a number of bandit methods to this framework, namely Thompson Sampling and Upper Confidence Bound, combined with novel extensions of $EC^2$ and IG which can handle tests with different and stochastic costs, which we call W-$EC^2$ and W-IG, respectively. Thereby, in addition to Thompson Sampling, we develop W-$EC^2$ and W-IG oracles utilizing a Bayesian variant of UCB (called BayesUCB by Kaufmann et al. 2012) that fits well to our framework and enables the UCB method to effectively employ prior knowledge in the form of a prior distribution.

4. We demonstrate the effectiveness of our framework via several experimental studies performed on data sets from a variety of important domains. We find that Thompson Sampling yields the best performance in most of the experiments we perform.

5. We theoretically analyze the performance of Thompson Sampling within our framework for cost-efficient online decision making and establish an upper bound on its Bayesian regret.

## 2 Related Work

**Cost-efficient information acquisition.** The primary goal of cost-efficient information acquisition (also called Bayesian active learning in the literature) algorithms is to sequentially select from a number of available costly tests to make a decision (such as making a prediction about a label) with a minimal cost. To minimize cost, these tests are performed until a sufficient level of confidence in the decision is reached. Prominent algorithms for cost-efficient information acquisition are Information Gain (IG) (originally developed by Lindley 1956), $EC^2$ (Golovin et al., 2010), and Uncertainty Sampling (US). Bayesian Active Learning by Disagreement (BALD) (Houlsby et al., 2011) proposes an equivalent formulation of the underlying utility function in IG which provides practical and computational advantages. $EC^2$ has been proved to enjoy adaptive submodularity (Golovin et al., 2010), and thus yields near-optimal cost. Acquisition functions in Bayesian optimization have also been used in active learning. Probability of Improvement (PI) (Kushner, 1964) and Expected Improvement (EI) (Jones et al., 1998) are two important acquisition functions used in

active learning. Both PI and EI use surrogate models (e.g., Gaussian Processes) to select the next sample to query (see Di Fiore et al. (2024) for more details). The works of Rahbar et al. (2023); Chen et al. (2017) consider active information acquisition in an online setting. The regret bounds in these works are based on formulating the problem as a Partially Observable Markov Decision Process (POMDP) and are *exponential* in the total number of possible tests. In this work, we consider a CMAB formulation of the problem and derive a regret bound that is *linear* in the number of tests (and *sublinear* w.r.t. the horizon $T$). In addition, unlike Rahbar et al. (2023); Chen et al. (2017), our framework supports variability in the cost of the tests depending on the outcomes of the tests and the decisions.

**Combinatorial semi-bandit algorithms.** CMAB methods have been utilized to address online learning problems in various settings but often without considering cost-efficient information acquisition. Durand & Gagné (2014) adapt a combinatorial variant of Thompson Sampling for *online feature selection* where the agent has a fixed budget for tests and the reward is a linear function of the test outcomes. Both assumptions are used in later works studying the problem from a CMAB perspective (Bouneffouf et al., 2017; 2021), but differ from the setting considered in this work (e.g., the budget for information acquisition is fixed with no concept of achieving a decision, and the costs are independent of the outcomes of the tests and the decisions). Wang & Chen (2018) analyze the regret of combinatorial Thompson sampling, but their proof technique is not directly applicable to our problem since they do not allow changes in the set of available super arms. Therefore, in this work, we adopt the approach originally developed by Russo & Van Roy (2014) for standard and linear MAB. Finally, it is notable that the analysis of Wang & Chen (2018) yields an instance dependent regret bound, whereas our analysis is focused on deriving Bayesian and instance-independent bounds.

**Online decision making.** Our work relates to the theoretical online decision making framework presented by Cesa-Bianchi et al. (2021). Their framework aims to maximize total Return on Investment (ROI) in a sequence of decision making tasks, where each task involves accepting or rejecting an innovation. The authors provide an algorithm to learn ROI-maximizing decision making policies, with theoretical guarantees of convergence to an optimal policy. In the framework, accept/reject decisions are made by drawing i.i.d. samples from a probability distribution modeling the value of an innovation. In contrast, our approach involves making low-cost decisions by performing a sequence of different tests (with different costs). Furthermore, our work is not limited to binary accept/reject decisions.

---

**Algorithm 1** Online decision making protocol

---
1: $\{x_1, x_2, \ldots, x_n\}$: available tests
   $[m]$: possible decisions
   $\mathcal{A} = \{a_{ij} | i \in [n], j \in [m]\}$: set of base arms
   $\mu_{ij}^{(0)}$ and $\mu_{ij}^{(1)}$: costs of base arm $a_{ij}$ (given test outcomes 0 and 1, respectively)
   $\theta_{ij} = Pr(x_i = 1 | y = j)$
2: **for** t = 1, 2, ..., T **do**
3:    Receive problem instance $\mathbf{x}^{(t)}$.
4:    **for all** selected tests $x_i^{(t)}$ (by an oracle) **do**
5:       Observe outcome of test $x_i^{(t)} \sim \text{Bernoulli}(\theta_{ij})$.
6:    **end for**
7:    Make a decision $y^{(t)} = j \in [m]$ based on test outcomes.
8:    Super arm $S^{(t)} = \{a_{ij} \in \mathcal{A} | \text{test } x_i^{(t)} \text{ is performed}\} \in \mathcal{I}^{(t)}$ where $\mathcal{I}^{(t)} \subseteq 2^{\mathcal{A}}$ is set of feasible super arms.
9:    Receive reward $R(S^{(t)})$ where $\mathbb{E}[R(S^{(t)})] = -\sum_{ij \in S^{(t)}} \mu_{ij}$ and $\mu_{ij} = \mu_{ij}^{(1)} \times \theta_{ij} + \mu_{ij}^{(0)} \times (1 - \theta_{ij})$.
10: **end for**

---

## 3 Problem Formulation

In this section, we propose our new formulation of the online decision making problem which incorporates the cost of acquiring test outcomes. In each time step $t$, we receive the problem instance (e.g., data point) $\mathbf{x}^{(t)}$ with $n$ tests $\{x_1^{(t)}, x_2^{(t)}, \ldots, x_n^{(t)}\}$. Here, we assume that all test outcomes (unknown until after each test

is performed) are binary (i.e., $x_i^{(t)} \in \{0, 1\}$ for all $i$), but our methods can easily be extended to other test values (see Section 6.3). The goal is to make an accurate decision $y^{(t)} \in [m]$ for that problem instance with a minimal cost of performing tests ($m$ represents the total number of options for the decision, or *outcomes* if the correct decision is viewed as a random variable), where we denote the correct decision by $y^*$. As an example, let's consider the medical problem of diagnosing lung cancer. At each time step $t$, a patient $\mathbf{x}^{(t)}$ arrives, and our goal is to make a decision (prediction) $y^{(t)} \in \{1(benign), 2(malignant), 3(infection)\}$ ($m = 3$). In this problem, there are several tests that can be performed to reach a decision. For instance, available tests can include a CT[1] scan, a PET[2] scan, genetic testing and a biopsy ($n = 4$). These tests can have positive or negative outcomes (test results), which will be used in the decision making process.

Consistent with the common Naïve Bayes assumption, we let all tests $x_i^{(t)}$ be mutually independent and Bernoulli-distributed conditional on the decision $y^{(t)}$. The Naïve Bayes assumption is common for sequential decision making, even in the offline setting, and is consistent with previous works including Chen et al. (2017); Rahbar et al. (2023). This assumption is needed for computing the gains in our approximate oracles defined in Section 4.1 (W-IG and W-EC$^2$). We also let $y^*$ be known given a full realization of $\mathbf{x}^{(t)}$. This implies that upon performing all available tests, we know the correct decision for the problem instance. This is realistic and consistent with the settings of Golovin et al. (2010); Chen et al. (2017); Rahbar et al. (2023), though they do not follow a CMAB formulation. As an example, consider a medical diagnosis problem. This property means that if we perform all available medical tests, we can determine the correct diagnosis.

Regarding the costs, we assume that they directly depend on the outcomes of the tests and the decisions. As motivation, again consider the medical diagnosis problem. Here, the costs (e.g., time and patient's discomfort) of medical tests can vary based on the underlying affliction or the outcome of the test itself. For example, in the case of diagnosing cancer through biopsy procedures, the cost of the biopsy can change based on the specific underlying condition and the outcome of the test. If the suspected cancerous tissue can be accessed easily and shows a positive result for cancer cells, the biopsy procedure may cause lower costs in terms of time and discomfort. Conversely, if the tissue is in a challenging area or requires multiple biopsies, the procedure's complexity may cause higher costs, both in terms of time and increased patient's discomfort.

We model the problem described above by an elegant design of a stochastic combinatorial multi-armed bandit (CMAB) problem (Cesa-Bianchi & Lugosi, 2012) with semi-bandit feedback. Briefly, CMAB corresponds to a reward maximization problem with a set of base arms. In each time step, the agent selects a subset of base arms. After playing this subset of base arms, called a *super arm*, the agent receives a reward from the environment. The objective is to maximize the expected sum of rewards within a time horizon $T$, which is typically formulated as a regret minimization task.

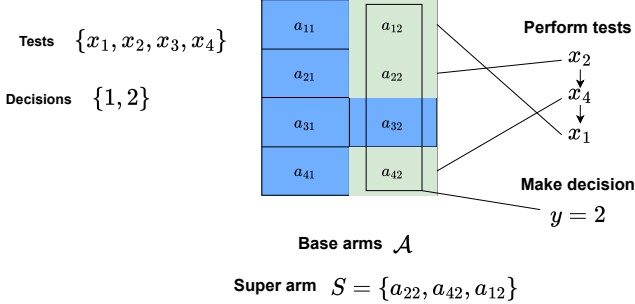

Figure 1: Illustration of base arms and super arms in our problem formulation. In this example, we have four tests ($n = 4$) and two possible decisions ($m = 2$). Based on these tests and decisions, we have $4 \times 2$ possible base arms $\mathcal{A} = \{a_{ij} | i \in [4], j \in [2]\}$. If we perform tests with indices $2, 4, 1$, and we make decision $y = 2$ (based on the test results), then the super arm will be $S = \{a_{22}, a_{42}, a_{12}\}$.

---

[1]Computed Tomography
[2]Positron Emission Tomography

In our problem, since the costs of tests depend on both test and decision outcomes, we need to consider a set of base arms that reflect both. Therefore, we define the set $\mathcal{A}$ of $n \times m$ base arms $a_{ij}$ ($i \in [n]$ and $j \in [m]$). In each time step $t$, the agent selects a super arm $S^{(t)} \in 2^{\mathcal{A}}$ and receives the reward $R(S^{(t)}) = R^{(t)}$. A base arm $a_{ij}$ in $S^{(t)}$ indicates that the test with index $i$ is performed for making decision $y^{(t)} = j$ (see Figure 1). We define $\mathcal{I}^{(t)} \subseteq 2^{\mathcal{A}}$ as the set of *feasible* super arms at time $t$ which yield a decision, i.e., each feasible super arm consists of tests and a single decision (concretely, if $a_{ij}, a_{lk} \in S \in \mathcal{I}^{(t)}$, then $j = k$). The feedback at time $t$ of a base arm $a_{ij} \in \mathcal{A}$ (observable by the agent if $a_{ij} \in S^{(t)}$) is the corresponding outcome of test $x_i^{(t)}$, which given a decision $y^{(t)} = j$ is Bernoulli-distributed (test outcomes are binary) with an unknown (to the agent) parameter $\theta_{ij} \triangleq Pr(x_i = 1 | y = j)$, for all $i, j$. While tests may, in practice, be selected sequentially, the super arm (and base arms) are only known *in hindsight* after the decision has been made. This is intentional and allows us to analyze the problem as a CMAB.

We let the cost (assumed to be fixed and known) of each base arm $a_{ij}$ be $\mu_{ij}^{(0)} \in [0, 1]$ and $\mu_{ij}^{(1)} \in [0, 1]$ (given test outcomes 0 and 1, respectively). Then, we define the expected reward of the selected feasible super arm $S^{(t)}$ as the negative sum of expected base arm costs, such that $\mathbb{E}[R(S^{(t)})] = -\sum_{ij \in S^{(t)}} \mu_{ij}$, where

$$\mu_{ij} \triangleq \mu_{ij}^{(1)} \times \theta_{ij} + \mu_{ij}^{(0)} \times (1 - \theta_{ij}), \tag{1}$$

with the vector of all $\mu_{ij}$'s denoted by $\boldsymbol{\mu}$ and the vector of all $\theta_{ij}$'s denoted by $\boldsymbol{\theta}$. We let $\boldsymbol{\mu}^*$ be the true underlying mean cost vector, with corresponding parameter vector $\boldsymbol{\theta}^*$. This is a novel problem; the previous studies are special cases of this problem with $\mu_{ij}^{(0)} = \mu_{ij}^{(1)} = c_i$, where $c_i$ is the cost of performing test $i$ and is independent of test outcome or the decision. We show an outline of our online decision making protocol in Algorithm 1.

We define the suboptimality gap at time $t$ of a feasible super arm $S$ as

$$\Delta_{\boldsymbol{\mu}^*}^{(t)}(S) \triangleq f_{\boldsymbol{\mu}^*}(S^{*,(t)}) - f_{\boldsymbol{\mu}^*}(S), \tag{2}$$

where $f_{\boldsymbol{\gamma}}(M) \triangleq (-\sum_{ij \in M} \gamma_{ij})$ denotes the expected reward of super arm $M$ given an arbitrary mean cost vector $\boldsymbol{\gamma}$, and $S^{*,(t)} \triangleq \operatorname{argmax}_{S \in \mathcal{I}^{(t)}} f_{\boldsymbol{\mu}^*}(S)$ is the optimal super arm found by the oracle applied on the feasible set $\mathcal{I}^{(t)}$. Then, the *regret* (which the agent should minimize) for a time horizon $T$ is defined as

$$\text{Regret}_{\boldsymbol{\mu}^*}(T) \triangleq \sum_{t \in [T]} \Delta_{\boldsymbol{\mu}^*}^{(t)}(S^{(t)}). \tag{3}$$

## 4 CMAB Methods for ODM

In our framework, we utilize *oracles* (i.e., optimization algorithms) which, given a parameter vector $\boldsymbol{\theta}$, output super arms with maximum expected reward (or approximately, in the case of approximate oracles), such that

$$Oracle(\boldsymbol{\theta}) \triangleq \operatorname*{argmax}_{S \in \mathcal{I}^{(t)}} \mathbb{E}\left[R(S) | \boldsymbol{\theta}\right]. \tag{4}$$

The $\theta_{ij}^*$ values of the true parameter vector $\boldsymbol{\theta}^*$ are initially unknown, but to be able to employ prior knowledge, we consider a prior distribution over $\theta_{ij}^*$'s. Formally, we assume that each $\theta_{ij}^*$ follows a Beta distribution, $\text{Beta}(\alpha_{ij}^{(0)}, \beta_{ij}^{(0)})$. The reason is that a Beta distribution is defined on range $[0, 1]$, and parameter of a Bernoulli distribution should be in this range (each $\theta_{ij}^*$ is parameter of a Bernoulli distribution). We also assume a fixed and known distribution over decision outcomes, denoted by $P(y)$ (i.e., the learning agent knows the marginal distribution *a priori*). However, even if the oracle is assumed to always be able to find the correct decision (i.e., it only needs to minimize the number of tests), the optimization problem is NP-hard, and thus in practice, we use approximate oracles (introduced in Section 4.1).

A *greedy* method to address the problem defined in Section 3 is that in each time step $t$, we use an oracle with the current (MAP) estimate of $\boldsymbol{\theta}^*$. However, this greedy method may converge to a sub-optimal super arm due to lack of exploration. To balance exploration and exploitation, we adapt Thompson Sampling and BayesUCB to our problem.

**Thompson Sampling.** Algorithm 2 summarizes our framework, with Thompson Sampling used as exploration method. In each time step $t$, we sample the parameters from the current posterior distribution (line 2). We then (in lines 3-4) use the *oracle* to get a super arm based on the sampled parameters, and perform the tests sequentially. Finally, we use Algorithm 3 to update the posterior distribution of the parameters. Algorithm 3 uses the fact that Beta distribution is conjugate prior to the Bernoulli distribution.[3]

**BayesUCB.** To utilize BayesUCB for exploration instead of Thompson Sampling, we have to modify lines 2-3 of Algorithm 2 such that instead of sampling parameters from the posterior distribution, we retrieve *optimistic* parameter estimates. Like Kaufmann et al. (2012) (done for classical bandits), we achieve this by utilizing quantiles of the posterior distribution. However, since neither gain function (Eq. 7 or Eq. 11) of the approximate oracles introduced in this section is monotonically increasing w.r.t. the provided parameter vector $\boldsymbol{\theta}$, a high value of a base arm parameter $\theta_{ij}$ does not necessarily mean that test $i$ is more likely to be selected. Hence, with $Q_\lambda(\eta)$ denoting the quantile function (such that under the distribution $\lambda$, $\Pr\left(X \le Q_\lambda(\eta)\right) = \eta$), we compute both an upper confidence bound $\overline{\theta}_{ij}^{(t)} = Q_{\text{Beta}(\alpha_{ij}^{(t)}, \beta_{ij}^{(t)})}(1 - 1/t)$ and a lower confidence bound $\underline{\theta}_{ij}^{(t)} = Q_{\text{Beta}(\alpha_{ij}^{(t)}, \beta_{ij}^{(t)})}(1/t)$ for each base arm $a_{ij}$, which we pass to the oracles. Then, while computing Eq. 7 and Eq. 11, whenever a high value for a term involving $\theta_{ij}$ makes the algorithm *more* likely to select test $i$, we use $\overline{\theta}_{ij}^{(t)}$ as an optimistic estimate, and correspondingly, $\underline{\theta}_{ij}^{(t)}$ for terms making $i$ *less* likely to be selected.

---

**Algorithm 2** Cost-efficient online decision making

---

**Require:** $P(y)$; $\boldsymbol{\mu}$; prior parameters $(\alpha_{ij}^{(0)}, \beta_{ij}^{(0)})$'s
1: **for** t = 1, 2, ..., T **do**
2:     Sample $\theta_{ij}^{(t)}$ from Beta$(\alpha_{ij}^{(t-1)}, \beta_{ij}^{(t-1)})$ for all $i, j$.
3:     $S^{(t)} \leftarrow Oracle(\boldsymbol{\theta}^{(t)})$ (e.g., Algorithm 4).
4:     Make decision based on $S^{(t)}$, and observe $x_i^{(t)}$ for all $i$ such that $a_{ij} \in S^{(t)}$.
5:     Observe correct decision $y^{(t)}$.
6:     Call Algorithm 3 and obtain Beta$(\alpha_{ij}^{(t)}, \beta_{ij}^{(t)})$ for all $i, j$.
7: **end for**

---

**Algorithm 3** Posterior update

---

**Require:** $S^{(t)}$; $x_i^{(t)}$ for all $i$ such that $a_{ij} \in S^{(t)}$; $y^{(t)} = k$; $\alpha_{ij}^{(t-1)}$'s and $\beta_{ij}^{(t-1)}$'s
1: **for** each $x_i^{(t)}$ **do**
2:     **if** $x_i^{(t)} = 1$ **then**
3:        $\alpha_{ik}^{(t)} \leftarrow \alpha_{ik}^{(t-1)} + 1$
4:     **else**
5:        $\beta_{ik}^{(t)} \leftarrow \beta_{ik}^{(t-1)} + 1$
6:     **end if**
7: **end for**

---

## 4.1 Cost-Efficient Approximate Oracles

Solving the optimization problem in Eq. 4 is intractable in general (see, e.g., Golovin et al. 2010; Chakaravarthy et al. 2007). Therefore, to implement line 3 of Algorithm 2, we propose two different *approximate oracles* using algorithms that greedily select tests to optimize a surrogate objective function, *Information Gain* (IG) (Zheng et al., 2012) and EC$^2$ (Golovin et al., 2010). We introduce extensions of IG and EC$^2$ to handle tests with stochastic costs, which we call *Weighted IG* (W-IG) and *Weighted EC$^2$* (W-EC$^2$) respectively. In both of these, all probabilities involved are computed using the fixed and known decision

---

[3]Since test outcomes are binary, then, given decision $y = j$, each test $x_i$ follows a Bernoulli distribution with parameter $\theta_{ij}$.

distribution $P(y)$ and the conditional distribution over test outcomes $P(\mathbf{x}|y)$ (determined by a given parameter vector $\boldsymbol{\theta}$, see line 3 in Algorithm 2). In particular, the dependence on $\boldsymbol{\theta}$ is omitted in the following section to simplify notation.

Before introducing W-IG and W-EC$^2$, we need to define the notions of *hypothesis* and *decision region*. We call a full realization of test outcomes (i.e., of all possible tests) a hypothesis. Therefore, with $n$ binary tests, we have $2^n$ different possible hypotheses. Let $\mathcal{H} \triangleq \{0,1\}^n$ be the set of all possible hypotheses for $n$ tests. We can partition $\mathcal{H}$ into $m$ disjoint sets. We call each of these disjoint sets a *decision region*. Each decision region corresponds to a decision $y \in [m]$. So our objective will be finding the correct decision region, for which the cost of performing tests is low. We want to emphasize that one important difference between these oracles and standard CMAB oracles is that they are *interactive*. In other words, during a single time step, they sequentially perform tests and observe outcomes to select subsequent tests and make a decision. To our knowledge, our work is the first work which utilizes this approach for CMABs.

**Weighted IG algorithm (W-IG).** In the standard IG algorithm, based on a set of previously observed test outcomes, we select the test that maximizes the reduction of entropy in the decision regions. Formally, the gain of a test $i$ in the IG algorithm is defined as:

$$\Delta_{\text{IG}}\left(i \mid \mathbf{x}_{\mathcal{P}}\right) \triangleq \mathbb{H}(y|\mathbf{x}_{\mathcal{P}}) - \mathbb{E}_{x_i|\mathbf{x}_{\mathcal{P}}}[\mathbb{H}(y|\mathbf{x}_{\mathcal{P}\cup\{i\}})], \tag{5}$$

where $\mathcal{P}$ is the set of previously performed tests, and $\mathbf{x}_{\mathcal{P}}$ is the vector of results of tests in $\mathcal{P}$. The random variable for decision regions is denoted $y$, and $\mathbb{H}(z)$ is the Shannon entropy of a random variable $z$. In the W-IG algorithm, we sequentially perform tests that maximize $\Delta_{\text{IG}}\left(i \mid \mathbf{x}_{\mathcal{P}}\right)$ divided with the expected cost of test $i$ given the previously performed tests. In other words, after performing and observing the outcomes of the tests in $\mathcal{P}$, the next test to perform will be

$$i^* = \underset{i \in ([n]\backslash\mathcal{P})}{\operatorname{argmax}} \Delta_{\text{W-IG}}\left(i \mid \mathbf{x}_{\mathcal{P}}\right), \tag{6}$$

where

$$\Delta_{\text{W-IG}}\left(i \mid \mathbf{x}_{\mathcal{P}}\right) \triangleq \frac{\Delta_{\text{IG}}\left(i \mid \mathbf{x}_{\mathcal{P}}\right)}{\text{Cost}(i)}, \tag{7}$$

and

$$\text{Cost}(i) \triangleq \sum_{j \in [m]} \sum_{q \in \{0,1\}} \mu_{ij}^{(q)} \times Pr\left(x_i = q, y = j \mid \mathbf{x}_{\mathcal{P}}\right), \tag{8}$$

given test outcome costs $\mu_{ij}^{(0)}$ and $\mu_{ij}^{(1)}$ for all $i, j$. We call the set of hypotheses $\mathcal{H}_i = \{h|P(x_i|h) > 0\}$ *consistent* with the outcome of test $i$. We continue performing tests until all hypotheses in $\bigcap_{i \in \mathcal{P}} \mathcal{H}_i$ belong to the same decision region, i.e., there is only one decision region remaining.

**Weighted EC$^2$ algorithm (W-EC$^2$).** The standard EC$^2$ algorithm starts with a graph with $\mathcal{H}$ as the set of nodes. We have an edge $(h, h')$ between two hypotheses $h$ and $h'$ if and only if they do not belong to the same decision region. The weight of an edge $(h, h')$ is set to $w_{hh'} = P(h \mid \mathbf{x}_{\mathcal{P}}) \times P(h' \mid \mathbf{x}_{\mathcal{P}})$ where $P(\cdot \mid \mathbf{x}_{\mathcal{P}})$ is the posterior distribution of a hypotheses after performing tests. Naturally, the weight of a set of edges $\mathcal{E}$ is defined as the sum of weights of edges in $\mathcal{E}$, i.e., $w(\mathcal{E}) \triangleq \sum_{(h,h')\in\mathcal{E}} w_{hh'}$. We say that performing a test $i$ *cuts* an edge $(h, h')$ if $h \notin \mathcal{H}_i$ or $h' \notin \mathcal{H}_i$. We denote the set of edges cut by test $i$ with $K(i)$.

We now define the objective function for the EC$^2$ algorithm as

$$f_{\text{EC}^2}\left(\mathcal{P}\right) \triangleq w\left(\bigcup_{i \in \mathcal{P}} K(i)\right). \tag{9}$$

Based on the objective function above, we define the EC$^2$ gain of a test $i$ as

$$\Delta_{\text{EC}^2}\left(i \mid \mathbf{x}_{\mathcal{P}}\right) \triangleq \mathbb{E}_{x_i|\mathbf{x}_{\mathcal{P}}}\left[f_{\text{EC}^2}\left(\mathcal{P} \cup \{i\}\right) - f_{\text{EC}^2}\left(\mathcal{P}\right)\right], \tag{10}$$

and, similar to W-IG, the W-EC$^2$ gain as

$$\Delta_{\text{W-EC}^2}\left(i \mid \mathbf{x}_{\mathcal{P}}\right) \triangleq \frac{\Delta_{\text{EC}^2}\left(i \mid \mathbf{x}_{\mathcal{P}}\right)}{\text{Cost}(i)}, \tag{11}$$

with $\text{Cost}(i)$ defined as in Eq. 8.

Also like with W-IG, in the W-EC$^2$ algorithm, we sequentially perform tests that maximize the gain $\Delta_{\text{W-EC}^2}\left(i \mid \mathbf{x}_{\mathcal{P}}\right)$. We stop performing tests when all edges are cut, which means that we have only one decision region left.

To be able to implement W-IG and W-EC$^2$ (calculate $\Delta_{\text{W-IG}}$ and $\Delta_{\text{W-EC}^2}$), similar to standard IG and EC$^2$, we assume that the outcomes of the tests are conditionally independent given the decision. We show the test selection process in W-IG and W-EC$^2$ algorithms in Algorithm 4.

---

**Algorithm 4** Cost-efficient approximate oracle

---

**Require:** $\text{Alg} \in \{\text{W-IG}, \text{W-EC}^2\}$; $P(y)$; $\boldsymbol{\mu}$; $\boldsymbol{\theta}$
 1: Enumerate hypotheses in $\mathcal{H}$
 2: $\mathcal{P} = \{\}$
 3: **while** more than one decision region left **do**
 4:    Select next test (with $\Delta_{\text{Alg}}(\cdot)$ computed w.r.t. $\boldsymbol{\theta}$) $i^* = \underset{i \in ([n] \backslash \mathcal{P})}{\text{argmax}} \Delta_{\text{Alg}}\left(i \mid \mathbf{x}_{\mathcal{P}}\right)$.
 5:    Perform test $i^*$ and observe $x_{i^*}$.
 6:    Update $P(h \mid \mathbf{x}_{\mathcal{P}})$ based on results of tests in $\mathcal{P}$.
 7: **end while**
 8: Select the decision associated to the remaining decision region.

---

## 5 Theoretical Analysis

In this section, we theoretically analyze the framework proposed in Section 4. Specifically, we provide an upper bound on the Bayesian regret (defined below) of the online decision making procedure in Algorithm 2. The analysis technique that we use extends the approach by Russo & Van Roy (2014) to our setting and yields an analysis which is significantly simpler than those by Chen et al. (2013) and Wang & Chen (2018). Furthermore, in our problem, due to the interactive nature of the oracle, the set of available arms changes in each time step. In general, a bandit problem with changing sets of available arms is called a *bandit with sleeping arms*, a setting which is not considered in the analyses of combinatorial UCB (CUCB) by Chen et al. (2013) and combinatorial Thompson sampling (CTS) by Wang & Chen (2018). Our analysis holds for interactive oracles through the sleeping arms assumption. Moreover, the analyses of CUCB and CTS yield instance-dependent regret bounds, whereas we focus on an instance-independent (and Bayesian) bound, which also requires a different type of analysis (e.g., a different regret decomposition). Furthermore, the analysis of CTS is tailored for one specific choice of uninformative prior, while ours, in principle, allows for an arbitrary selection of the prior distribution (whether uninformative or informative).

In our problem setting, the set of feasible super arms $\mathcal{I}^{(t)}$ is restricted such that $y^{(t)} = y^*$ (i.e., correct decisions are guaranteed, and the objective is to minimize the cost of tests). This means that a super arm is feasible if and only if the test outcomes determine the correct decision region. [4] We also consider access to a perfect *oracle* which can solve the optimization problem Eq. 4. As mentioned earlier, the problem under consideration is NP-hard. On the other hand, assuming access to an exact oracle is common for analyses of regret bounds for CMABs, even in settings where the exact oracle needs to deal with an NP-hard problem. For instance, the work of Hüyük & Tekin (2020) assumes access to exact oracles for NP-hard network problems. Åkerblom et al. (2023) consider the intractable problem of stochastic bottleneck identification in an online setting. In fact, via the elegant CMAB formulation of our problem (in contrast to, e.g., the POMDP formulation of Rahbar et al. 2023 and Chen et al. 2017), we are able to obtain a regret bound that is linear in the number of tests (rather than exponential).

---

[4]We relax this assumption in the experiments of Section 6.4.

The Bayesian regret is defined as the expected value of Regret($T$) (see Eq. 3), with the expectation taken over the prior distribution of $\boldsymbol{\mu}^*$ and other random variables (e.g., randomness in the rewards and the policy of the agent). Formally,

$$\text{Bayesian Regret}(T) \triangleq \mathbb{E}\left[\sum_{t \in [T]} \Delta_{\boldsymbol{\mu}^*}^{(t)}(S^{(t)})\right]. \tag{12}$$

We then prove the following theorem which establishes an upper bound on the Bayesian regret of Thompson Sampling applied to our framework.

**Theorem 5.1.** *The Bayesian Regret of Algorithm 2 is $\mathcal{O}(mn\sqrt{T \log T})$.*

*Proof.* We begin our regret analysis by a decomposition of $\mathbb{E}[\Delta_{\boldsymbol{\mu}^*}^{(t)}(S^{(t)})]$ in the following lemma.

**Lemma 5.2.** *For any upper confidence bound $U^{(t)} : \mathcal{I}^{(t)} \to \mathbb{R}$ and lower confidence bound $L^{(t)} : \mathcal{I}^{(t)} \to \mathbb{R}$,*

$$\mathbb{E}[\Delta_{\boldsymbol{\mu}^*}^{(t)}(S^{(t)})] = \mathbb{E}[U^{(t)}(S^{(t)}) - L^{(t)}(S^{(t)})] + \mathbb{E}[L^{(t)}(S^{(t)}) - f_{\boldsymbol{\mu}^*}(S^{(t)})] + \mathbb{E}[f_{\boldsymbol{\mu}^*}(S^{*,(t)}) - U^{(t)}(S^{*,(t)})]. \tag{13}$$

*Proof.* This lemma is based on Proposition 1 of Russo & Van Roy (2014). Given the history of played super arms (and base arms) and their rewards, we know that in Thompson Sampling the optimal super arm $S^{*,(t)}$ and the played super arm $S^{(t)}$ follow the same distribution. Since $U^{(t)}(\cdot)$ is a deterministic function, we conclude that given the history, the expected values of $U^{(t)}(S^{(t)})$ and $U^{(t)}(S^{*,(t)})$ are equal, and we have:

$$\mathbb{E}[\Delta_{\boldsymbol{\mu}^*}^{(t)}(S^{(t)})] = \mathbb{E}[U^{(t)}(S^{(t)}) - f_{\boldsymbol{\mu}^*}(S^{(t)})] + \mathbb{E}[f_{\boldsymbol{\mu}^*}(S^{*,(t)}) - U^{(t)}(S^{*,(t)})],$$

and by adding and subtracting $\mathbb{E}[L^{(t)}(S^{(t)})]$ we prove Lemma 5.2. □

Additionally, we define lower and upper confidence bounds as

$$L^{(t)}(S) \triangleq f_{\hat{\boldsymbol{\mu}}^{(t-1)}}(S) - \sum_{ij \in S} \sqrt{\frac{2 \log T}{N^{(t-1)}(ij)}} \tag{14}$$

$$U^{(t)}(S) \triangleq f_{\hat{\boldsymbol{\mu}}^{(t-1)}}(S) + \sum_{ij \in S} \sqrt{\frac{2 \log T}{N^{(t-1)}(ij)}} \tag{15}$$

where $\hat{\mu}_{ij}^{(t)}$ is the average cost of base arm $a_{ij}$ till time $t$, and $N^{(t)}(ij)$ is the number of times base arm $a_{ij}$ has been played till time $t$. We continue the proof by bounding each term in Lemma 5.2. For the second and third terms, we have the following lemma (where the proof is in Appendix B.1).

**Lemma 5.3.** $\mathbb{E}[L^{(t)}(S^{(t)}) - f_{\boldsymbol{\mu}^*}(S^{(t)})] \leq \frac{2mn}{T}, \mathbb{E}[f_{\boldsymbol{\mu}^*}(S^{*,(t)}) - U^{(t)}(S^{*,(t)})] \leq \frac{2mn}{T}.$

In order to bound the first term of the regret decomposition, we utilize the following lemma (where the proof uses a similar technique as Lemma 8 by Åkerblom et al. (2023) for the combinatorial batched feedback setting (with batch size 1). See Appendix B.3).

**Lemma 5.4.** $\sum_{t \in [T]} \mathbb{E}[U^{(t)}(S^{(t)}) - L^{(t)}(S^{(t)})] \leq 2mn\sqrt{8T \log T}.$

Now, based on Lemma 5.2, Lemma 5.3, and Lemma 5.4 we have Bayesian Regret($T$) $\leq 4mn + 2mn\sqrt{8T \log T} = \mathcal{O}(mn\sqrt{T \log T})$. □

We utilize the structure of the problem in our analysis and thereby, the bound derived in Theorem 5.1 is significantly tighter than the ones by Rahbar et al. (2023); Chen et al. (2017) which take a different perspective than our CMAB formulation. Our bound is also consistent with Bayesian upper bounds for combinatorial semi-bandit methods developed in other (standard) settings.

# 6 Experiments

In this section, we empirically validate our cost-efficient online decision making framework via various experiments [5]. Unless otherwise specified, we consider Beta(2,2) as the prior distribution of all $\theta_{ij}^*$'s. We employ both of the cost-efficient approximate oracles introduced in Section 4.1 (W-IG and W-EC$^2$) as the oracle in Algorithm 2. We also employ the hypothesis enumeration procedure proposed by Chen et al. (2017) to generate the most likely hypotheses for each decision region. In Section 6.3, we provide an extension of our framework to real-valued non-binary test outcomes. Moreover, In Section 6.4, we demonstrate the applicability of our framework in the setting where the decision regions for the set of hypotheses are not known. We run each experiment with 5 different random seeds on a single-node CPU machine with 8 cores, 16GB memory and macOS. We implement the algorithms in Python mainly using NumPy (Harris et al., 2020), NetworkX (Hagberg et al., 2008) and SciPy (Virtanen et al., 2020).

**Data sets.** We apply our framework to the LED display domain data set from the UCI repository (Breiman et al., 1988). Additionally, we use the ProPublica recidivism (Compas) data set (Larson et al., 2016) and the Fair Isaac (Fico) credit risk data set (FICO et al., 2018), which both are preprocessed in the same way as by Hu et al. (2019). The aforementioned data sets contain relatively large numbers of tests and few decision outcomes, which correspond well to, e.g., the medical diagnosis example in Section 1. In applications like the charging station selection example, it might instead be reasonable to expect a large number of decision outcomes (individual charging stations or groups sharing certain characteristics) and relatively few tests (questions). Hence, in order to thoroughly evaluate the latter type of setting, we create a synthetic data set (called Navigation) with the desired characteristics. With $n = 5$ tests and $m = 20$ decision outcomes, we sample parameters $\theta_{ij}^*$ from the prior distribution Beta(2,2) for each $i \in [n]$ and $j \in [m]$, which subsequently use to generate the data set. Moreover, in Section 6.2, we present the application of our framework to a real-world troubleshooting case study. Finally, to demonstrate the applicability of our methods to real-valued test outcomes, we investigate the Breast Cancer Wisconsin data set (Street et al., 1993) (Section 6.3).

**Algorithms.** In our experiments, we investigate the cost of decision making using both W-IG and W-EC$^2$ algorithms embedded in our framework (Algorithm 2). For each algorithm, we investigate both Thompson Sampling and BayesUCB for exploration. In addition to Thompson Sampling (TS) and BayesUCB (BUCB), we consider the following baselines for selecting tests in each time step:

**Random Information Acquisition.** Within our framework, we can use a random subset of tests decision making. This baseline continues performing random tests until only one decision region is left (similar to W-IG and W-EC$^2$).

**All.** This baseline always performs all available tests, and does not consider the cost.

**DPP.** Determinantal Point Process (DPP) provides a way to select diverse sets. With this method, we use exact sampling (Derezinski et al., 2019) (finite DPP with likelihood kernel) to subsample columns (tests). We use the implementation of Gautier et al. (2019). To perform the sampling, in each time step $t$, we provide all data points (all test results for all data points) until time $t - 1$ to the sampler. The number of selected tests is related to the number of eigenvectors of the kernel matrix.

Table 1: Average cost ($\pm$ standard deviation) of decision making in a time step (over the entire learning period).

| Data set \ Algorithm | W-EC$^2$-TS | W-EC$^2$-BUCB | W-IG-TS | W-IG-BUCB | Random | All | DPP |
|---|---|---|---|---|---|---|---|
| Navigation | $1.746 \pm 0.030$ | $1.746 \pm 0.030$ | $1.859 \pm 0.020$ | $1.867 \pm 0.023$ | $1.975 \pm 0.008$ | $2.305 \pm 0.000$ | $2.300 \pm 0.002$ |
| LED | $2.032 \pm 0.053$ | $2.061 \pm 0.059$ | $2.418 \pm 0.015$ | $2.438 \pm 0.019$ | $2.644 \pm 0.025$ | $3.147 \pm 0.000$ | $3.137 \pm 0.002$ |
| Fico | $1.159 \pm 0.003$ | $1.101 \pm 0.002$ | $1.935 \pm 0.075$ | $2.067 \pm 0.064$ | $3.078 \pm 0.005$ | $8.353 \pm 0.000$ | $8.249 \pm 0.056$ |
| Compas | $3.056 \pm 0.009$ | $3.090 \pm 0.017$ | $4.197 \pm 0.039$ | $4.034 \pm 0.028$ | $4.719 \pm 0.006$ | $6.101 \pm 0.000$ | $5.923 \pm 0.049$ |
| Breast cancer | $1.437 \pm 0.019$ | $1.554 \pm 0.032$ | $2.091 \pm 0.048$ | $2.213 \pm 0.136$ | $6.376 \pm 0.087$ | $14.181 \pm 0.000$ | $9.286 \pm 0.077$ |
| Troubleshooting | $6.207 \pm 0.013$ | $7.615 \pm 0.034$ | $12.010 \pm 0.078$ | $12.563 \pm 0.117$ | $23.236 \pm 0.072$ | $38.113 \pm 0.000$ | $24.538 \pm 0.309$ |

---

[5]The source code for our experiments can be accessed here on GitHub.

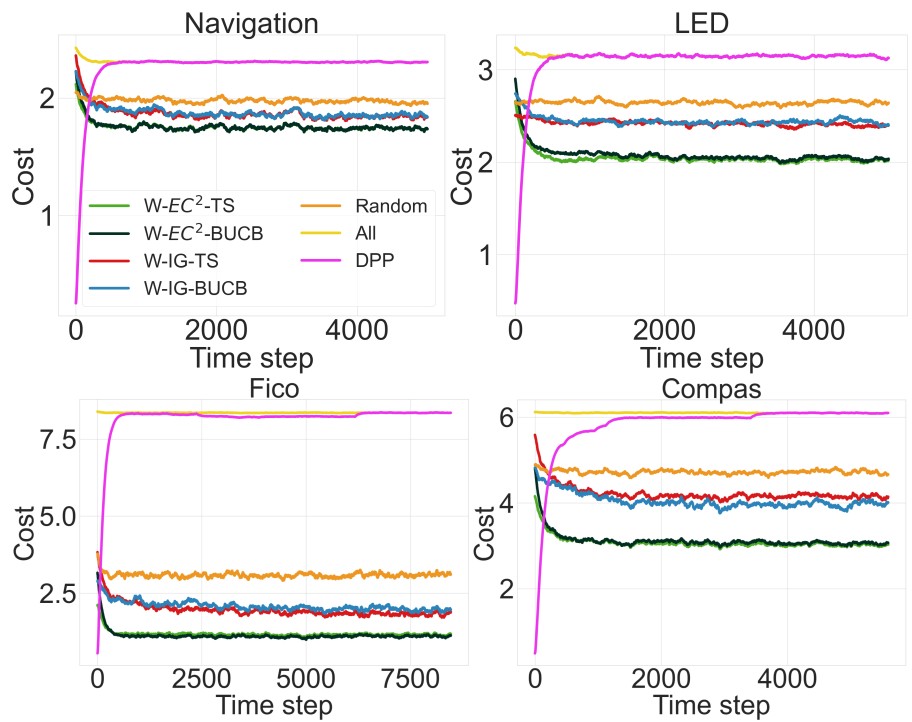

Figure 2: Decision making costs of our framework applied to four different data sets.

## 6.1 Experimental Results

We first investigate the four data sets Navigation, LED, Fico, and Compas. As mentioned, to guarantee accurate decision regions for all hypotheses, we implement the algorithm outlined by Chen et al. (2017) for enumeration of hypotheses. In short, this algorithm produces the most probable hypotheses based on a decision and the conditional probabilities of test outcomes associated with that decision. To assign the correct decisions to hypotheses, we extract the parameter vector $\boldsymbol{\theta}$ from the entire data set and subsequently utilize the enumeration procedure to generate the most likely hypotheses for each decision. However, the parameters are not known to the agent after the enumeration of hypotheses. We assign fixed costs $\mu_{ij}^{(0)} \in [0, 1]$ and $\mu_{ij}^{(1)} \in [0, 1]$ (randomly sampled from a uniform distribution) for each data set before performing our experiments.

In Figure 2, we illustrate the cost[6] of performing tests (i.e., $-\mathbb{E}[R(S^{(t)})]$) during online learning for different algorithms. The cost of each test is calculated based on Eq. 1, where the parameter $\theta_{ij}$ is derived from the data. The (maximum) numbers of enumerated hypotheses per decision region are 2, 5, 70 and 70 for Navigation, LED, Fico and Compas datasets, respectively. Our results in Figure 2 show that our framework yields the lowest information acquisition cost for all data sets. DPP has a low cost initially since it only uses a few data points to perform sampling, but after a few time steps its cost becomes as high as 'All'. During these initial time steps DPP can make incorrect decisions. Additionally, we observe that the W-EC$^2$ algorithm always has the lowest cost. This result is consistent with the theoretical results of Golovin et al. (2010) for the cost of the original EC$^2$ algorithm in the offline setting. We also observe that the exploration methods (TS and BayesUCB) exhibit very similar performances in terms of cost. During online learning, it is observed that the cost of the random algorithm remains relatively constant, whereas the costs associated with the W-IG and W-EC$^2$ algorithms decrease until they converge to a low value. This behavior is attributed to the utilization of parameter estimates by the W-IG and W-EC$^2$ algorithms when selecting tests. As the knowledge of the agent about these parameters improves over the course of learning, the associated costs

---

[6]The costs reported here directly correspond to the (instant) regret up to only a constant term. Therefore, the order of the methods and their relative performances remain the same.

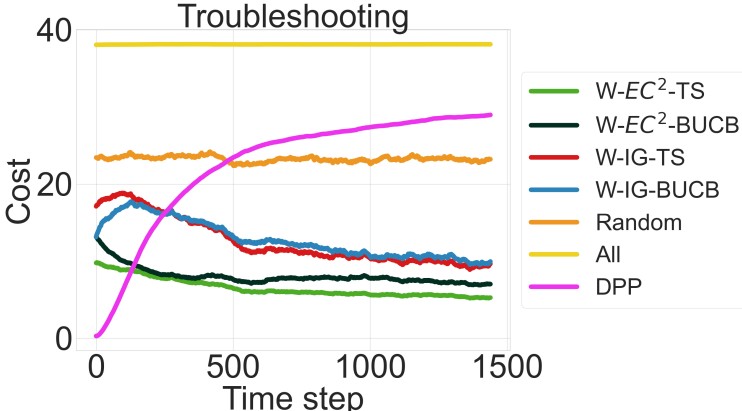

Figure 3: Decision making costs of our framework when applied to online troubleshooting.

decline. Furthermore, in Table 1, we report the average cost of decision making in a single time step for different data sets. It is obvious that our framework incurs significantly lower costs for performing tests.

## 6.2 Application to Online Troubleshooting

In this section, we study the application of our decision making framework for online troubleshooting. The data set for this *real-world* application is collected from contact center agents. The agents solve problems from mobile devices. We use a subset of this data set with 15 possible decisions and 74 tests. Each test corresponds to a symptom that a customer may or may not see on the device.

We experiment on approximately 1500 different scenarios (i.e., roughly 100 scenarios for each decision). A scenario starts with a customer entering the system. Then the online learning agent aims at finding the correct decision by asking a sequence of multiple questions of the customer (performing tests). The goal for the agent is to ask the most informative questions in a cost-efficient manner. We enumerate 15 hypotheses for each decision. Figure 3 shows the cost of making decisions for different time steps (i.e., scenarios) in the same setting as Section 6.1. The results show that our framework enables the agent to find the correct decisions for troubleshooting in mobile devices with a significantly lower cost compared to other methods. Similar to previous observations, the W-EC$^2$ algorithm remains the most cost-effective method within our framework.

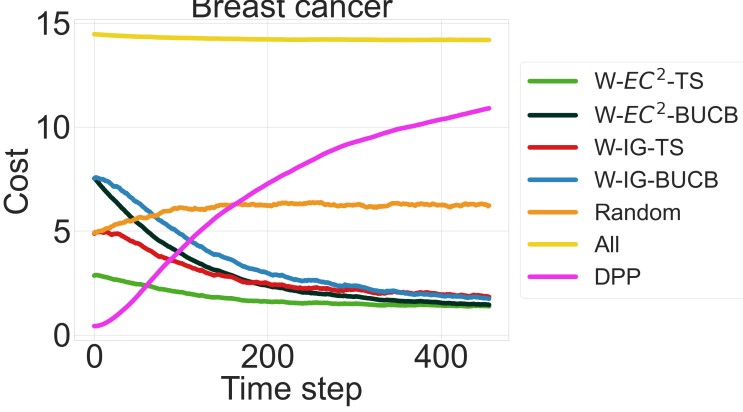

Figure 4: Decision making costs of our framework extended to real-valued test outcomes in the Breast Cancer Wisconsin data set.

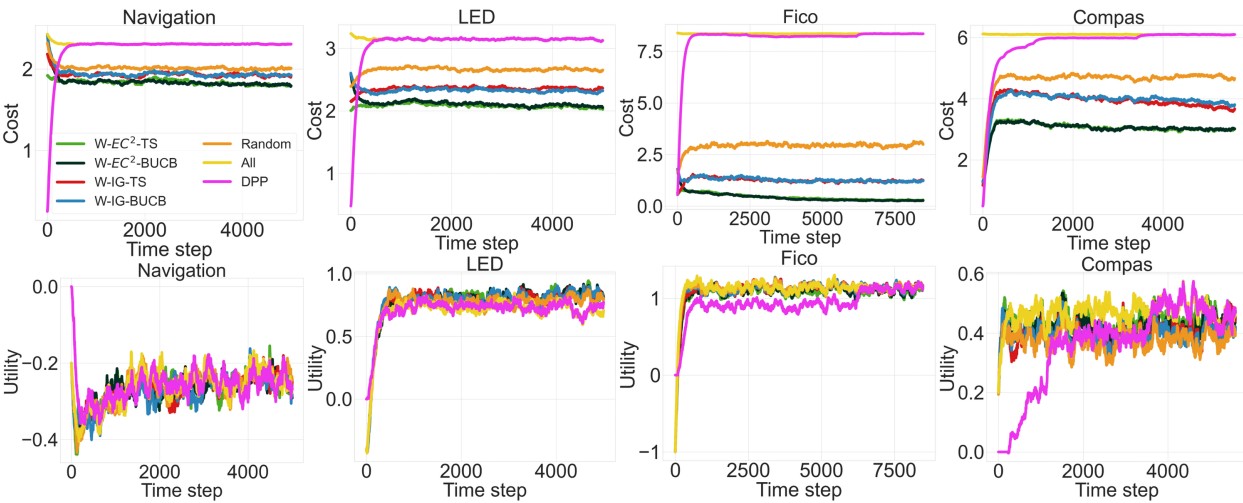

Figure 5: Decision making cost (first row) and utility (second row) of our framework applied to the setting that the decision regions are not known beforehand.

## 6.3 Extension to Real-Valued Test Outcomes

In this section, we extend our experimental results to real-valued (non-binary) test outcomes. For this purpose, we adapt the discretization method proposed by Rahbar et al. (2023). Specifically, for each test, we consider different thresholds for "binarization". Then, in each time step, we can calculate the gain ($\Delta_{\text{W-IG}}$ or $\Delta_{\text{W-EC}^2}$) for the tests using different binarization thresholds and choose the one with a maximal gain. We employ this method on the Breast Cancer Wisconsin data set. In this data set, we predict the diagnosis from 30 real-valued medical tests computed from an image of a fine needle aspirate. Figure 4 illustrates the decision making cost incurred by our framework when applied to this data set. We enumerate 70 hypotheses per decision region. Similar to the results depicted in Figure 2, our framework consistently achieves the lowest cost, demonstrating its effectiveness in this setting. In particular, we observe that W-EC²-TS yields the best results, and W-IG-TS, W-EC²-BUCB, and W-IG-BUCB are other suitable methods for this task.

## 6.4 Extension to Unknown Decision Regions

In this section, to demonstrate the generality of our framework, we examine the setting where the decision regions for the enumerated hypotheses are not known. To assign hypotheses to decision regions, we employ the hypothesis enumeration algorithm discussed in Section 6.1 to generate the most probable hypotheses for each decision region based on the current parameter sample. Therefore, each hypothesis is assumed to correspond to the decision it is assigned to. This situation may lead to incorrect decisions by the agent. Specifically, in the context of online learning, our objectives are: i) to achieve highly accurate decisions, and ii) to minimize the cost of performing tests. To quantify the accuracy of the decisions, we employ a utility function. Specifically, we assign a utility of 2 for a correct decision and $-1$ for an incorrect one. Additionally, we consider an "unknown" decision when the correct hypothesis is assigned to multiple decision regions. We define the utility of such decisions as 0.

The first row in Figure 5 shows the cost of making decisions for different data sets (when the correct decisions of the hypotheses are not known). We use the same numbers of enumerated hypotheses as Figure 2. Similar to the findings in Figure 2, we observe that our framework (with both W-IG and W-EC²) yields a final decision with significantly lower cost than other algorithms. Again, we observe that EC² (with both Thompson Sampling and BayesUCB) has the lowest cost of performing tests. The IG algorithm results in the second lowest cost, and the random algorithm for picking tests is the third option.

As previously mentioned, when faced with unknown decision regions, our goal is to enhance the accuracy of our decisions during online learning. The second row in Figure 5 illustrates the utility achieved by various information acquisition algorithms. Notably, our framework demonstrates comparable performance to both All and DPP in the early stages of learning. Therefore, our approach proves capable of making precise decisions with a significantly reduced cost. We note that DPP yields very low costs in the early time steps, but the corresponding utility is also very low.

## 7    Conclusion

We propose a novel framework for online decision making based on an elegant design of a combinatorial multi-armed bandit problem, which incorporates the cost of performing tests, and where the costs may be stochastic and depend on both test and decision outcomes. Within this framework, we develop various cost-efficient online decision making methods such as W-EC$^2$ and W-IG. We also adapt Thompson Sampling and BayesUCB, methods that are commonly used for exploration. In particular, we provide a theoretical upper bound for the Bayesian regret of Thompson Sampling. We demonstrate the performance of the framework on a number of data sets from different domains.

**Acknowledgments**

The work of Arman Rahbar and Morteza Haghir Chehreghani was partially supported by the Wallenberg AI, Autonomous Systems and Software Program (WASP) funded by the Knut and Alice Wallenberg Foundation. The work of Niklas Åkerblom was partially funded by the Strategic Vehicle Research and Innovation Programme (FFI) of Sweden, through the project EENE (reference number: 2018-01937).

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

## A  Nomenclature

Table 2: Table of notation used in this paper.

| Notation | Description |
|---|---|
| $t$ | Time step |
| $\mathbf{x}$ | A problem instance |
| $\mathbf{x}^{(t)}$ | A problem instance received at time $t$ |
| $i$ | Test index |
| $x_i$ | Test with index $i$ |
| $x_i^{(t)}$ | Test with index $i$ performed at time $t$ |
| $y$ | A decision |
| $y^{(t)}$ | A decision made at time $t$ |
| $n$ | Number of available tests |

Table 2: Table of notation used in this paper.

| Notation | Description |
|---|---|
| $m$ | Number of possible decisions |
| $y^*$ | Correct decision |
| $T$ | Time horizon |
| $a_{ij}$ | Base arm related to test with index $i$ and decision with index $j$ |
| $\mathcal{A}$ | Set of base arms |
| $S$ | A super arm |
| $S^{(t)}$ | A super arm selected at time $t$ |
| $R^{(t)}$ | Reward received at time $t$ |
| $R(S)$ | Reward of super arm $S$ |
| $\mathcal{I}^{(t)}$ | Set of feasible super arms at time $t$ |
| $\theta_{ij}$ | Parameter of Bernoulli distribution for test $i$ and decision $j$ |
| $\mu_{ij}^{(0)}$ | Cost of base arm $a_{ij}$ when test outcome is 0 |
| $\mu_{ij}^{(1)}$ | Cost of base arm $a_{ij}$ when test outcome is 1 |
| $\mu_{ij}$ | Expected cost of base arm $a_{ij}$ |
| $\boldsymbol{\mu}$ | Vector of all $\mu_{ij}$'s |
| $\boldsymbol{\theta}$ | Vector of all $\theta_{ij}$'s |
| $\boldsymbol{\mu}^*$ | True mean cost vector |
| $\boldsymbol{\theta}^*$ | True parameter vector |
| $\Delta_{\boldsymbol{\mu}^*}^{(t)}(S)$ | Suboptimality gap at time $t$ of super arm $S$ with mean cost vector is $\boldsymbol{\mu}^*$ |
| $f_{\boldsymbol{\mu}}(S)$ | Expected reward of super arm $S$ given a mean cost vector $\boldsymbol{\mu}$ |
| $S^{*,(t)}$ | Optimal super arm at time $t$ |
| $\mathrm{Regret}_{\boldsymbol{\mu}^*}(T)$ | Regret for a time horizon $T$ when true mean cost vector is $\boldsymbol{\mu}^*$ |
| $\alpha_{ij}^{(t)}$ | First shape parameter of Beta distribution related to $\theta_{ij}$ at time $t$ |
| $\beta_{ij}^{(t)}$ | Second shape parameter of Beta distribution related to $\theta_{ij}$ at time $t$ |
| $Q_\lambda(\eta)$ | Quantile function (if X has distribution $\lambda$, then $\Pr\left(X \leq Q_\lambda(\eta)\right) = \eta$) |
| $\overline{\theta}_{ij}^{(t)}$ | Upper confidence bound for $\theta_{ij}$ used in BayesUCB |
| $\underline{\theta}_{ij}^{(t)}$ | Lower confidence bound for $\theta_{ij}$ used in BayesUCB |
| $\mathcal{H}$ | Set of all possible hypotheses |
| $h$ | A hypothesis |
| $\mathcal{P}$ | Set of performed tests |
| $\mathbf{x}_{\mathcal{P}}$ | Vector of results of tests in $\mathcal{P}$ |
| $\mathbb{H}(z)$ | Shannon entropy of a random variable $z$ |
| $\Delta_{\mathrm{Alg}}(i \mid \mathbf{x}_{\mathcal{P}})$ | Gain of test $i$ with algorithm Alg when $\mathcal{P}$ is the set of performed tests |
| $i^*$ | Index of next test to perform |
| $\mathcal{H}_i$ | Set of hypotheses consistent with the outcome of test $i$ |
| $(h, h')$ | An edge between two hypotheses $h$ and $h'$ |
| $w_{hh'}$ | Weight of an edge $(h, h')$ |
| $\mathcal{E}$ | A set of edges |
| $w(\mathcal{E})$ | Weight of a set of edges $\mathcal{E}$ |
| $K(i)$ | Set of edges cut by test $i$ |
| $f_{\mathrm{EC}^2}(\mathcal{P})$ | Objective function of EC$^2$ for a set of performed tests $\mathcal{P}$ |
| $\mathrm{Bayesian\ Regret}(T)$ | Bayesian regret over a time horizon $T$ |
| $U^{(t)}(S)$ | Upper confidence bound for a super arm $S$ |
| $L^{(t)}(S)$ | Lower confidence bound for a super arm $S$ |
| $\hat{\mu}_{ij}^{(t)}$ | Average cost of base arm $a_{ij}$ till time $t$ |
| $N^{(t)}(ij)$ | Number of times base arm $a_{ij}$ has been played till time $t$ |
| $\bar{x}_{ij}^{(\tau)}$ | Average of first $\tau$ observations of test $i$ under decision $j$ |

# B    Additional Proofs

## B.1    Proof of Lemma 5.3

*Proof.*

$$\mathbb{E}[L^{(t)}(S^{(t)}) - f_{\boldsymbol{\mu}^*}(S^{(t)})]$$

$$= \mathbb{E}\left[ - \sum_{ij \in S^{(t)}} \hat{\mu}_{ij}^{(t-1)} - \sum_{ij \in S^{(t)}} \sqrt{\frac{2 \log T}{N^{(t-1)}(ij)}} + \sum_{ij \in S^{(t)}} \mu_{ij}^* \right]$$

$$= \sum_{ij \in S^{(t)}} \mathbb{E}\left[ - \hat{\mu}_{ij}^{(t-1)} - \sqrt{\frac{2 \log T}{N^{(t-1)}(ij)}} + \mu_{ij}^* \right]$$

$$\leq \sum_{ij \in S^{(t)}} \mathbb{E}\left[ \left| \mu_{ij}^* - \hat{\mu}_{ij}^{(t-1)} \right| - \sqrt{\frac{2 \log T}{N^{(t-1)}(ij)}} \right]$$

$([z]^+ = \max\{0, z\})$

$$\leq \sum_{ij \in \mathcal{A}} \mathbb{E}\left[ \left[ \left| \mu_{ij}^* - \hat{\mu}_{ij}^{(t-1)} \right| - \sqrt{\frac{2 \log T}{N^{(t-1)}(ij)}} \right]^+ \right]$$

$$= \sum_{ij \in \mathcal{A}} \left( \mathbb{E}\left[ \left| \mu_{ij}^* - \hat{\mu}_{ij}^{(t-1)} \right| - \sqrt{\frac{2 \log T}{N^{(t-1)}(ij)}} \right| \text{``bad event''} \right] \times Pr\{\text{``bad event''}\} \right), \tag{16}$$

where "bad event" is $\left| \mu_{ij}^* - \hat{\mu}_{ij}^{(t-1)} \right| > \sqrt{\frac{2 \log T}{N^{(t-1)}(ij)}}$.

The last equality comes from the fact that $\left[ \left| \mu_{ij}^* - \hat{\mu}_{ij}^{(t-1)} \right| - \sqrt{\frac{2 \log T}{N^{(t-1)}(ij)}} \right]^+ = 0$ if $\left| \mu_{ij}^* - \hat{\mu}_{ij}^{(t-1)} \right| \leq \sqrt{\frac{2 \log T}{N^{(t-1)}(ij)}}$.
We continue with the following lemma.

**Lemma B.1.** $Pr\left\{ \exists t \in [T] \; \exists ij \in \mathcal{A}, \left| \mu_{ij}^* - \hat{\mu}_{ij}^{(t-1)} \right| > \sqrt{\frac{2 \log T}{N^{(t-1)}(ij)}} \right\} \leq \frac{2}{T}$.

*Proof.* To prove Lemma B.1, we use Eq. 1 which implies that $\left| \mu_{ij}^* - \hat{\mu}_{ij}^{(t-1)} \right| = \left| \mu_{ij}^{(1)} - \mu_{ij}^{(0)} \right| \left| \theta_{ij}^* - \hat{\theta}_{ij}^{(t-1)} \right|$ together with the fact that $\left| \mu_{ij}^{(1)} - \mu_{ij}^{(0)} \right| \leq 1$, as well as Hoeffding's inequality. The full proof is given in Appendix B.2. □

Continuing from Eq. 16, using Lemma B.1, and noting that $\left| \mu_{ij}^* - \hat{\mu}_{ij}^{(t-1)} \right| \leq 1$ we have

$$\mathbb{E}[L^{(t)}(S^{(t)}) - f_{\boldsymbol{\mu}^*}(S^{(t)})] \leq \sum_{ij \in \mathcal{A}} \frac{2}{T} = \frac{2mn}{T}. \tag{17}$$

We can similarly show that

$$\mathbb{E}[f_{\boldsymbol{\mu}^*}(S^{*,(t)}) - U^{(t)}(S^{*,(t)})] \leq \frac{2mn}{T}. \tag{18}$$

□

## B.2 Proof of Lemma B.1

*Proof.* Let $\bar{x}_{ij}^{(\tau)}$ be the average of first $\tau$ observations of $x_i$ under the decision region $y = j$. We have:

$$Pr\left\{\exists t \in [T]\ \exists ij \in \mathcal{A}, \left|\mu_{ij}^* - \hat{\mu}_{ij}^{(t-1)}\right| > \sqrt{\frac{2\log T}{N^{(t-1)}(ij)}}\right\}$$

$$\overset{Eq.(1)}{=} Pr\left\{\exists t \in [T]\ \exists ij \in \mathcal{A}, \left|\mu_{ij}^{(1)} - \mu_{ij}^{(0)}\right|\left|\theta_{ij}^* - \hat{\theta}_{ij}^{(t-1)}\right| > \sqrt{\frac{2\log T}{N^{(t-1)}(ij)}}\right\}$$

(since $\left|\mu_{ij}^{(1)} - \mu_{ij}^{(0)}\right| \leq 1$)

$$\leq Pr\left\{\exists t \in [T]\ \exists ij \in \mathcal{A}, \left|\theta_{ij}^* - \hat{\theta}_{ij}^{(t-1)}\right| > \sqrt{\frac{2\log T}{N^{(t-1)}(ij)}}\right\}$$

(Union bound)

$$\leq \sum_{t\in[T]} \sum_{ij\in\mathcal{A}} \sum_{\tau=1}^{(t-1)} Pr\left\{\left|\theta_{ij}^* - \bar{x}_{ij}^{(\tau)}\right| > \sqrt{\frac{2\log T}{\tau}}\right\}$$

(assuming $T \geq |\mathcal{A}|$)

$$\leq T^3 \times 2T^{-4} = \frac{2}{T}.$$

The last inequality is due to Hoeffding's inequality (e.g., Theorem A.1 of Slivkins et al. (2019) with $\alpha = 2$).

$\square$

## B.3 Proof of Lemma 5.4

*Proof.*

$$\sum_{t\in[T]} \mathbb{E}[U^{(t)}(S^{(t)}) - L^{(t)}(S^{(t)})]$$

$$= \sum_{t\in[T]} \mathbb{E}\left[2 \sum_{ij\in S^{(t)}} \sqrt{\frac{2\log T}{N^{(t-1)}(ij)}}\right]$$

$$\leq \sqrt{8\log T} \sum_{t\in[T]} \mathbb{E}\left[\sum_{ij\in S^{(t)}} \frac{1}{\sqrt{N^{(t-1)}(ij)}}\right]$$

$$= \sqrt{8\log T} \sum_{ij\in\mathcal{A}} \mathbb{E}\left[\sum_{t:ij\in S^{(t)}} \frac{1}{\sqrt{N^{(t-1)}(ij)}}\right]$$

$$= \sqrt{8\log T} \sum_{ij\in\mathcal{A}} \mathbb{E}\left[\sum_{l\in[N^{(T)}(ij)]} \frac{1}{\sqrt{l}}\right]$$

(Lemma 1 of Russo & Van Roy 2014)

$$\leq 2\sqrt{8\log T} \sum_{ij\in\mathcal{A}} \mathbb{E}\left[\sqrt{N^{(T)}(ij)}\right]$$

(Cauchy–Schwarz inequality)

$$\leq 2\sqrt{8\log T}\,\mathbb{E}\left[\sqrt{mn\sum_{ij\in\mathcal{A}} N^{(T)}(ij)}\right]$$

$$\leq 2\sqrt{8\log T}\,\mathbb{E}\left[\sqrt{(mn)(mn)T}\right]$$

$$= 2mn\sqrt{8T\log T}.$$

$\square$

