# OpenReview forum: "Cost-Efficient Online Decision Making: A Combinatorial Multi-Armed Bandit Approach"
_TMLR — Accepted by TMLR_

### Review · Reviewer_JNrp · 2024-10-28

**Summary Of Contributions:**

The authors propose a framework for an online decision-making problem where decisions are made based on the results of cost-incurring tests. They utilize the Combinatorial Multi-Armed Bandit (CMAB) framework, specifically Thompson Sampling (TS) and BayesUCB-based algorithms, to estimate the test-decision parameters. Additionally, they information acquisition-based methods to select which tests to perform based on the estimated parameters. The authors also provide a sublinear upper bound for the Bayesian regret and present simulation results demonstrating the performance of their approach.

**Audience:**

Yes

**Claims And Evidence:**

Yes

**Requested Changes:**

1. As stated in the weaknesses section, the assumption that "a super arm is feasible if and only if the test outcomes determine the correct decision region" is very limiting. Could the authors please justify this?
2. Could the authors compare their chosen oracles to some other active-learning algorithms? For instance Expected Improvement (EI), Probability of Improvement (PI), or Bayesian Active Learning by Disagreement (BALD).
3. Could the authors provide a real-world example of their proposed framework, where they define tests, decisions, costs, etc.?
4. Could the authors provide guarantees/error bounds for the proposed oracles (W-IG and W-EC$^2$)?

**Strengths And Weaknesses:**

### Strengths
1. Combining CMAB with (cost-aware) information acquisition-based methods for solving cost-aware decision-making problems is an interesting approach.
2. Theoretical guarantees on the regret, albeit straightforward and not novel, are nice to have.
3. Simulations show that the authors' approach does better than some baseline algorithms on semi-synthetic datasets.

### Weaknesses
1. Adaptations of the TS and BayesUCB algorithms for parameter estimation lack novelty.
2. The framework relies on the Naïve Bayes assumption of conditional independence among tests given the decision.
3. The framework assumes that a super arm is feasible if and only if the test outcomes determine the correct decision region. This implies that the model focuses solely on minimizing test costs. Such an assumption may limit the applicability of the framework in settings where certain decisions naturally incur higher test costs. For instance, in a medical diagnosis scenario involving both flu and cancer, the current approach would consistently select lower-cost tests for flu, potentially neglecting the necessary, higher-cost tests required to accurately diagnose cancer.
4. There are theoretical results, but only for the Bayesian regret (i.e., CMAB part), and no guarantees for the oracles (W-IG and W-EC$^2$).

---

> ### Author Response · Authors · 2024-11-18
>
> We thank Reviewer **JNrp** for the constructive feedback. In what follows, we answer the questions raised by the reviewer.
>
> **1. Adaptations of the TS and BayesUCB algorithms for parameter estimation lack novelty.**
>
> Our aim is not to develop novel algorithms for multi-armed bandits in combinatorial settings; rather, our goal is to formulate the challenging problem of cost-efficient online decision making with the framework of combinatorial multi-armed bandits through a novel and elegant approach of defining base and super arms. Please refer to the end of the introduction section for a comprehensive list of our contributions.
>
> **2. The framework relies on the Naïve Bayes assumption of conditional independence among tests given the decision.**
>
> As stated in the paper, the Naïve Bayes assumption is common for sequential decision making, even in the offline setting. This assumption is needed for computing the gains in our approximate oracles (W-IG and W-EC$^2$) and is consistent with previous works including [1] and [2] (we have added this note in the revised version of the paper).
>
> **3. The framework assumes that a super arm is feasible if and only if the test outcomes determine the correct decision region. This implies that the model focuses solely on minimizing test costs. Such an assumption may limit the applicability of the framework in settings where certain decisions naturally incur higher test costs. For instance, in a medical diagnosis scenario involving both flu and cancer, the current approach would consistently select lower-cost tests for flu, potentially neglecting the necessary, higher-cost tests required to accurately diagnose cancer.**
>
> In the analyzed setting, the oracle will always give the correct decision (cancer in the example). So, in the flu/cancer example, the oracle always performs the lowest cost sequence of tests necessary to diagnose cancer.
>
> We relax this assumption in our experiments. As demonstrated in Section 6.4, our framework still works well in practice even without this assumption. Specifically, in the experiments of Section 6.4, we utilize our framework to simultaneously make highly accurate decisions and minimize the cost of performing tests. We emphasize that, as mentioned in Section 4, the optimization problem of Eq. 4 is intractable and challenging even with this assumption.
>
> **4. There are theoretical results, but only for the Bayesian regret (i.e., CMAB part), and no guarantees for the oracles (W-IG and W-EC$^2$).**
>
> While the optimization problem in Eq. 4 is NP-hard, the EC$^2$ and IG algorithms both follow an iterative and adaptive procedure that approximately solve it (in the deterministic costs setting). Notably, theoretical guarantees can be established on the performance of the adaptive and incremental algorithm EC$^2$. We refer the reviewer to [3] for further details. Extending these theoretical guarantees of approximate oracles to our stochastic costs setting is an interesting direction for future work.
>
> **References**
>
> [1] Rahbar, A., Ye, Z., Chen, Y. and Chehreghani, M.H. (2023). Efficient online decision tree learning with active feature acquisition. International Joint Conference on Artificial Intelligence (IJCAI '23).
>
> [2] Chen, Y., Renders, J., Chehreghani, M.H. and Krause, Andreas. (2017). Efficient Online Learning for Optimizing Value of Information: Theory and Application to Interactive Troubleshooting. Conference on Uncertainty in Artificial Intelligence.
>
> [3] Golovin, D., Krause, A. and Ray, D. (2010). Near-optimal Bayesian active learning with noisy observations. Conference on Neural Information Processing Systems.

---

> ### Author Response · Authors · 2024-11-18
>
> **Requested Changes:**
>
> **1. As stated in the weaknesses section, the assumption that "a super arm is feasible if and only if the test outcomes determine the correct decision region" is very limiting. Could the authors please justify this?**
>
> We addressed this question in our response to weakness 3. We added a note in the revised version of the paper.
>
> **2. Could the authors compare their chosen oracles to some other active-learning algorithms? For instance Expected Improvement (EI), Probability of Improvement (PI), or Bayesian Active Learning by Disagreement (BALD).**
>
> The BALD algorithm provides a reformulation of the IG utility function that has computational and practical advantages.
>
> EI and PI are not directly applicable to our setting. Both EI and PI are acquisition functions commonly used in Bayesian optimization and have been adapted in active learning as well. In both algorithms, a surrogate model (usually Gaussian processes) is used to select the next sample to query. PI uses likelihood of improvement (in the solution of the optimization problem) to select the next sample to query, while EI considers both likelihood and magnitude of improvement in sample selection.
>
> We added citations to these algorithms in the revised version of the paper.
>
> **3. Could the authors provide a real-world example of their proposed framework, where they define tests, decisions, costs, etc.?**
>
> In the revised version of the paper, we included more details in the medical diagnosis example.
>
> **4. Could the authors provide guarantees/error bounds for the proposed oracles (W-IG and W-EC$^2$)?**
>
> We addressed this question in our response to weakness 4.

---

### Review · Reviewer_ZRLU · 2024-11-08

**Summary Of Contributions:**

This paper presents a novel framework for cost-efficient online decision-making by leveraging a combinatorial multi-armed bandit (CMAB) approach. The core contributions can be summarized as follows:

- Problem Formulation: The authors introduce a new problem formulation for outcome-dependent, cost-efficient online decision-making, where tests and decisions incur stochastic costs. This framework generalizes traditional multi-armed bandit problems to include combinatorial and cost-efficient aspects.

- CMAB Framework: The proposed combinatorial semi-bandit framework involves selecting a subset of actions (referred to as "super arms") under combinatorial constraints, thus optimizing the trade-off between information gain and decision cost.

- Algorithmic Development: The authors adapt classical bandit methods, particularly Thompson Sampling and Bayesian Upper Confidence Bound (BayesUCB), and extend existing active information acquisition methods (like EC2 and Information Gain) to handle variable costs. The new algorithms, named W-EC2 and W-IG, optimize test selections to minimize overall decision costs.

- Theoretical Analysis: The paper establishes a theoretical upper bound on the Bayesian regret for Thompson Sampling, demonstrating the efficiency of their approach in the CMAB setting.

Experiments complement the theoretical analysis.

**Audience:**

Yes

**Broader Impact Concerns:**

No.

**Claims And Evidence:**

Yes

**Requested Changes:**

- Implement the visual aids requested in Weakness 2.

- Add the protocol as described in Weakness 3.

- (Optional but strongly recommended) Add a (near-)matching lower bound to complement Theorem 5 (Weakness 4.)

- ROI Maximization in Stochastic Online Decision-Making (Cesa-Bianchi et al., 2021) seems related. Please, add it to the related work section, and compare/contrast.

**Strengths And Weaknesses:**

**Strengths**

1. Novel Formulation: The combination of cost-efficient decision-making with combinatorial bandit algorithms is a significant extension of existing online learning models.

2. Theoretical Rigor: The paper provides formal theoretical guarantees, in the form of a regret upper bound.

**Weaknesses**

1. Heavy Notation: The paper introduces complex notation that can be overwhelming, especially in the problem formulation and algorithmic sections, where the reader is just getting acquainted with it.

2. Readability: The presentation is not easy to follow due to the notation and the way the material is presented. It would help to get some visual aids, such as representing the base arms as a matrix with a highlighted subset representing a superarm, and arrows pointing to the other quantities (e.g., the corresponding tests) to clarify the interactions between tests and decisions.

3. Presentation of the protocol: It would be helpful to have a fully detail pseudocode representing the protocol (similar to Algorithm 1, but where all notation is clarified). It was not immediate to me to get everything from the drawn-out description in Section 3. Place this float at the beginning of the problem formulation.

4. Missing matching lower bound

---

> ### Author Response · Authors · 2024-11-18
>
> We thank Reviewer **ZRLU** for the constructive feedback. In what follows, we answer the questions raised by the reviewer.
>
> **1. Heavy Notation: The paper introduces complex notation that can be overwhelming, especially in the problem formulation and algorithmic sections, where the reader is just getting acquainted with it.**
>
> We acknowledge that the notation introduced in our problem formulation can sometimes be complex, but this complexity is necessary to formulate the problem as a CMAB problem. As requested by the reviewer, we will make the notation clearer by adding visual aids in the revised version of the paper. We also added a table of notation to the appendix.
>
> **2. Readability: The presentation is not easy to follow due to the notation and the way the material is presented. It would help to get some visual aids, such as representing the base arms as a matrix with a highlighted subset representing a superarm, and arrows pointing to the other quantities (e.g., the corresponding tests) to clarify the interactions between tests and decisions.**
>
> We implemented the visual aids in the revised version of the paper.
>
> **3. Presentation of the protocol: It would be helpful to have a fully detail pseudocode representing the protocol (similar to Algorithm 1, but where all notation is clarified). It was not immediate to me to get everything from the drawn-out description in Section 3. Place this float at the beginning of the problem formulation.**
>
> We added the protocol in the revised version of the paper.
>
> **4. Missing matching lower bound**
>
> There is a lower bound in [1] for the Bayesian regret of  standard stochastic bandits which is consistent with the dependency on  $\sqrt{T}$ in our upper bound. However, for combinatorial bandits, we are not aware of any existing lower bound for Bayesian regret (in the general setting).
>
> **Requested Changes:**
>
> **1. Implement the visual aids requested in Weakness 2.**
>
> We implemented the visual aids in the revised version of the paper.
>
> **2. Add the protocol as described in Weakness 3.**
>
> We added the protocol in the revised version of the paper.
>
> **3. (Optional but strongly recommended) Add a (near-)matching lower bound to complement Theorem 5 (Weakness 4.)**
>
> Please refer to our response to weakness 4.
>
> **4. ROI Maximization in Stochastic Online Decision-Making (Cesa-Bianchi et al., 2021) seems related. Please, add it to the related work section, and compare/contrast.**
>
> We added the mentioned paper to our related work section and compared it with our approach.
>
> Their framework aims to maximize total Return on Investment (ROI) in a sequence of decision making tasks, where each task involves accepting or rejecting an innovation. The authors provide an algorithm to learn ROI-maximizing decision making policies, with theoretical guarantees of convergence to an optimal policy. In the framework, accept/reject decisions are made by drawing i.i.d. samples from a probability distribution modeling the value of an innovation. In contrast, our approach involves making low-cost decisions by performing a sequence of different tests (with different costs). Furthermore, our work is not limited to binary accept/reject decisions.
>
> **References**
>
> [1] Lattimore, T., & Szepesvári, C. (2020). Bandit algorithms. Cambridge University Press.

---

### Review · Reviewer_ZvvS · 2024-11-11

**Summary Of Contributions:**

This paper formulates the online decision making problem as a combinatorial multi-armed bandits problem. The reward in the literature corresponds to the cost of each decision making. The paper proposes oracles to solve the combinatorial offline optimazation problem and propose a Thompson sampling as well as a Bayes UCB algorithm to solve the online setting. The paper provide both theoretical guarantees and empirical verifications for the proposed methods.

**Audience:**

Yes

**Claims And Evidence:**

Yes

**Requested Changes:**

Please see the weakness part. Section 3 should be modified for better readability.

**Strengths And Weaknesses:**

Strength:
1. It is reasonable to consider the cost of decision making problem and formulate as a combinatorial MAB problem.
2. The results are complete with both theoretical and empirical verifications.

Weakness:
1. It is not easy to follow the story of the work. In Section 3, what does the $m$ mean? The authors say that $m$ represents the total number of options for the decision. Can $m$ be regarded as a combination over $n$ tests thus each decision making corresponds to a selection over $n$ tests? If so, the base arm set size in the CMAB should be $n$ instead of $n\times m$.
2. In the end of page 3, "We let the cost (assumed to be fixed and known)", should "know" be changed as "unknown"? And what does test outcome mean? It seems that the paper assumes the cost to be known but the test outcome to be unknown. Can authors give a motivating example for this assumption?

---

> ### Author Response · Authors · 2024-11-18
>
> We thank Reviewer **ZvvS** for the constructive feedback. In what follows, we answer the questions raised by the reviewer.
>
> **1. It is not easy to follow the story of the work. In Section 3, what does the m mean? The authors say that m represents the total number of options for the decision. Can m be regarded as a combination over  n tests thus each decision making corresponds to a selection over n tests? If so, the base arm set size in the CMAB should be n instead of n×m.**
>
> To make our problem formulation more clear, we added a decision making protocol (Algorithm 1 in the revised version) and some visual aids (in Figure 1), as suggested by reviewer ZRLU. Additionally, we added a table of notations to the appendix.
>
> In our problem formulation, m corresponds to the number of possible decisions that we can make after performing a sequence of n tests. Our set of base arms needs to reflect both selected tests and the final decision because the costs depend on both. As mentioned in the paper, the selected super arm (and base arms) are only known in hindsight after the decision has been made. This CMAB formulation is non-trivial (it e.g., requires an elegant way of defining base arms and super arms), and is what allows us to derive regret bounds which are linear in the number of tests (in contrast to prior works [1,2]).
>
> Please let us know if any further clarification is needed.
>
> **2. In the end of page 3, "We let the cost (assumed to be fixed and known)", should "know" be changed as "unknown"? And what does test outcome mean? It seems that the paper assumes the cost to be known but the test outcome to be unknown. Can authors give a motivating example for this assumption?**
>
> As mentioned in Section 3 (problem formulation), the costs depend on the outcomes of the tests and the decisions (see the medical diagnosis example in Section 3). This problem is novel; previous studies address special cases where the cost of each test is independent of its outcome or the decision made. By test outcome, we refer to the result of performing a test. For instance, in the medical diagnosis example, the outcome of a biopsy procedure can be either negative or positive. This outcome is not known at the time we receive the problem instance; it becomes known only after performing the test. However, the costs associated with each test outcome and decision are known beforehand. For example, in the medical diagnosis problem, an expert can estimate the cost (including time and patient discomfort) of a biopsy procedure for a specific underlying affliction (decision) and a test result (negative or positive).
>
> Please let us know if any further clarification is needed.
>
> **References**
>
> [1] Rahbar, A., Ye, Z., Chen, Y. and Chehreghani, M.H. (2023). Efficient online decision tree learning with active feature acquisition. International Joint Conference on Artificial Intelligence (IJCAI '23).
>
> [2] Chen, Y., Renders, J., Chehreghani, M.H. and Krause, Andreas. (2017). Efficient Online Learning for Optimizing Value of Information: Theory and Application to Interactive Troubleshooting. Conference on Uncertainty in Artificial Intelligence.

---

### Review · Reviewer_rkex · 2024-11-11

**Summary Of Contributions:**

The paper addresses the problem of cost efficient online Multi-Armed Bandits exploration, i.e. when there is a cost attached to running the experiments. After a brief overview of related work, the paper formally introduces the CMAB formulation of the problem. Given the difficulty of the optimization used, it also provides two approximate oracles to solve it. It then provides some theoretical insights on the Bayesian Regret of the framework, before validating the approach on benchmark datasets and a case study on online troubleshooting.

**Audience:**

Yes

**Claims And Evidence:**

Yes

**Requested Changes:**

See weaknesses section. In short:
- Review clarity of notation, and comprehensively define everything
- Extend theoretical analysis to the approximate oracles

**Strengths And Weaknesses:**

Strengths:
- New formulation for cost efficiency as a CMAB is interesting and results look promising
- Theoretical analysis and bound on Bayesian Regret show good mathematical grounding for the idea
- Approximate oracles are elegant approaches to solve the untractable optimization problem
- Extensive empirical study of the methods in practice, both on benchmark datasets and the case study on online troubleshooting

Weaknesses:
- Lack of clarity sometimes in the reasoning and notations - e.g. modeling choices are not always justified (for instance, why specifically a a Beta distribution as a prior for the parameters in section 4?)
- No theoretical analysis of the approximate oracles, which for me is fairly significant - this is necessary to give insights on the practicality of the method (not just the framework which is not directly applicable)

---

> ### Author Response · Authors · 2024-11-18
>
> We thank Reviewer **rkex** for the constructive feedback. In what follows, we answer the questions raised by the reviewer.
>
> **1. Lack of clarity sometimes in the reasoning and notations - e.g. modeling choices are not always justified (for instance, why specifically a Beta distribution as a prior for the parameters in section 4?)**
>
> To improve clarity, we added some visual aids (Figure 1) as well as a decision making protocol (Algorithm 1), as suggested by reviewer ZRLU. We also included a notation table to the appendix.
>
> We added additional notes to justify our modeling choices. For instance, Beta prior distribution is used to model the parameter of a Bernoulli distribution because the parameter must be in range [0,1].
>
> Please let us know if further clarification is needed.
>
> **2. No theoretical analysis of the approximate oracles, which for me is fairly significant - this is necessary to give insights on the practicality of the method (not just the framework which is not directly applicable)**
>
> EC$^2$ and IG algorithms approximately solve the NP-hard optimization problem in Eq. 4 in an iterative and adaptive procedure (in the deterministic costs setting). There are theoretical guarantees on the performance of the adaptive and incremental algorithm EC$^2$. Please see [1] for further details. Extending these theoretical guarantees of approximate oracles to our stochastic costs setting is an interesting direction for future work.
>
> **Requested Changes:**
>
> **See weaknesses section. In short:**
>
> **Review clarity of notation, and comprehensively define everything**
>
> Please see our response to weakness 1.
>
> **Extend theoretical analysis to the approximate oracles**
>
> Please see our response to weakness 2.
>
> **References**
>
> [1] Golovin, D., Krause, A. and Ray, D. (2010). Near-optimal Bayesian active learning with noisy observations. Conference on Neural Information Processing Systems.

---

### Author Response · Authors · 2024-11-18

We appreciate the reviewers' feedback and have revised our paper accordingly. The updated version includes changes highlighted in blue text. Below, we address the questions and comments from each reviewer individually.

---

### Decision · Action_Editor_Bw8u · 2024-12-20

**Recommendation:** Accept as is

**Comment:**

The reviewers agree that the problem is relevant and that the formulation of online decision making as a combinatorial multi armed bandit problem is meaningful and novel.
The techniques employed in the design and analysis of the proposed algorithm are rather standard, but the results are comprehensive and sound.
The authors have already implemented the main suggestions of the reviewers so I deem the latest revision ready for publication.

**Audience:**

The proposed framework can be of interest for the community.

**Claims And Evidence:**

The claims are supported by convincing evidence.